# Effects of Sports Massage on the Physiological and Mental Health of College Students Participating in a 7-Week Intermittent Exercises Program

**DOI:** 10.3390/ijerph18095013

**Published:** 2021-05-10

**Authors:** Chih-Chien Shen, Yi-Han Tseng, Meng-Chun Susan Shen, Hsiao-Hsien Lin

**Affiliations:** 1Institute of Physical Education and Health, Yulin Normal University, 1303 Jiaoyu East Rd., Yulin 537000, China; g169168@gmail.com; 2Department of Tourism Leisure and Health Management, Chung Chou University of Science and Technology, No. 6, Lane 2, Sec. 3, Shanjiao Rd., Yuanlin City 510, Taiwan; yihjyu@gmail.com; 3Department of Business Administration, Asia University, Taichung 41354, Taiwan; kesalan@gmail.com; 4Department of Leisure Industry Management, National Chin-Yi University of Technology, Taichung 41170, Taiwan

**Keywords:** sports massage, body composition, physiological health, mental health, college students

## Abstract

The purpose of the research is to analyze the improvement in the physical and mental health of college students after intermittent exercises are performed by massage. The present study employed a mixed research method. An experimental study was conducted to analyze the current status of the volunteers’ sports performance and body composition, and then a questionnaire was designed for the subjects’ physical and mental health. The data were then analyzed using SPSS 26.0 software for statistical analysis such as *t*-test and ANOVA. The subjects were then interviewed to collect their opinions on the study results, and finally, the results were explored by multivariate analysis. The study found that intermittent exercise can help university students develop physical fitness and performance, improve body composition, and regulate physical and mental health. The combination of intermittent exercise with sports massages further enhanced the performance of sit-ups and standing long jump, improve blood pressure, BMI, and self-confidence, as well as reducing suicidal tendencies (experimental group > control group). However, intermittent exercise participants still experienced fatigue, headache, emotional loss, and fear of depression, and the addition of sports massage did not significantly improve flexibility and cardiorespiratory endurance (control group > experimental group).

## 1. Introduction

Human resources are one of the main driving forces of a country’s economic development, and the young population is the cornerstone and hope for the country’s future development. With the advancement of technology, economic development, and medical quality, the average life expectancy of Chinese people has improved dramatically [1], providing stable human resources for national development. Due to the booming business and trade, the economic level has risen, the quality of life and the consumption ability have increased, the nutritional intake is convenient and diversified. However, due to the fast pace of life and the pressure of work, the obese population is increasing year by year, the number of patients with hypertension and cardiovascular diseases remains high [2], the age of patients is decreasing year by year [3,4], and the risk for young people is increasing [5], which depletes medical resources and affects public health and national development [6].

According to previous data, more than 70% of adolescents worldwide face the problem of low activity level and lack physical adaptability [7]. In addition, the increased environmental risk in the community due to the COVID-19 epidemic from 2019 onwards has decreased the willingness of adolescents and children to get out and exercise [8]. Physiologically, changes in lifestyle behaviors, increased sedentary or lying-down time [9], or reduced willingness to engage in exercise can lead to increased body weight, decreased cardiopulmonary and physical adaptability [10], as well as physiological symptoms such as disorders of muscles, bones, and perceptual nerves, and gastrointestinal discomfort [11,12]. Psychologically, the lack of social and interpersonal interaction and communication can easily lead to emotional instability, anxiety, panic, and increased psychological stress [10,13], resulting in adaptive disorders that may induce cardiovascular disease [14] and permanent vascular tension disorder under long-term deterioration [11]. It is evident that, in addition to the current health problems arising from low activity levels and inadequate physical adaptability [7,15], living under the COVID-19 environmental risk and response protocols poses a deeper negative impact on the well-being and growth of adolescents [8,13,14,15,16,17,18].

Studies have confirmed that the COVID-19 environment has significant effects on the mental or physical health of adolescents [8] which has become a major concern for governments [6,19,20]. It has also been shown that physical exercise can help improve physical and mental health [21,22], regulate psychological stress [23,24], and improve life and academic performance [25]. However, in fact, university students’ willingness to exercise is still decreasing due to requirement differences in the exercise environment, equipment and methods, convenience of exercise participation process [26], perception differences in the goal of physical appearance improvement, and comfort of exercise experience, as well as time and cost factors [27]. Therefore, making exercise decisions that require less time and cost and have significant exercise outcomes can reduce barriers to exercise participation, meet the objectives of adolescents’ exercise needs, and help improve their current physical and mental health problems [8,9,10,11,12,13,14].

Based on the intensity design, intermittent exercise can be divided into two modes: HICT and MICT. It is a break from the conventional method of long-duration low-intensity aerobic exercise for fat loss [28]. Both exercise intensities are gradually adopted by the public because they both improve cardiorespiratory fitness with short time requirements and high efficiency [29]. Although studies have shown that HICT exercise mode is more effective in improving well-being and physical fitness [30], long-term use of HICT training is associated with periodic fatigue [16]. Therefore, to promote the use of intermittent exercise to improve the physical and mental health of university students, periodic fatigue [16] will be an influential factor in the willingness of university students to exercise.

Exercise can improve well-being [22], but exercise fatigue can also cause negative effects on physical and mental health [28,29,30]. Soreness or fatigue after exercise will affect sleep quality and daily life [31] and affect participants’ willingness to exercise or performance [32]. Therefore, if intermittent exercise is recommended to improve physical fitness for health, finding ways to resolve or relieve fatigue and restore exercise performance [31,33,34,35] will be instrumental to the willingness of university students to engage in intermittent exercise.

Massage is one of the age-old tools of medical care in China [36,37]. Through massage techniques, it helps to relieve muscle tension, cardiovascular and visual disorders, and improve health [38], which has been validated in relevant studies [39]. Massage techniques should have the same effect in relieving fatigue produced by intermittent exercise. Therefore, the investigators believed that the combination of sports massage techniques would improve the fatigue in university students adopting intermittent exercise mode, which would help to further increase their willingness to engage in intermittent exercise.

Studies have shown that exercise can help improve health [22] and massage can relieve fatigue [38,39]. However, most of the current studies on intermittent exercise have focused on human health [38], analyzed exercise patterns [29], and investigated their effects on improving exercise performance [38] and fat loss [28]. Massage studies, on the other hand, focus on the treatment of cervical nerve [36] and lower back pain [39] as well as the relief of muscle soreness [40,41]. In the context of the current COVID-19 epidemic, only a few studies have examined the effects of massage and intermittent exercise on adolescents’ exercise behavior [42], daily activity and respiratory status [43,44], and physical and mental health [45,46]. Besides, there has been no study combining the themes of massage and intermittent exercise. Therefore, we believed that it would be helpful to fill the gap in the current research by examining the effects of massage on athletic performance, body composition, and perception of physical and mental health of exercise participants, based on the example of university students, and provide guidance for governments, educational institutions, teachers, and individual students.

## 2. Literature Review

### 2.1. Seven-Minute High-Intensity Circuit Training (HICT)

Intermittent exercise is a break from the norm to perform long periods of low-intensity aerobic exercise for fat loss [28]. According to the difference in exercise intensity, intermittent exercise can be subdivided into HICT and MICT, both of which have an improvement effect on the cardiorespiratory fitness of healthy adults. HICT is more effective, but both have the characteristics of short duration and high efficiency, which are gradually being adopted by the public [29].

The 7-min high-intensity circuit training (7 min HICT) exercise mode emphasizes that participants can exercise in a safe environment, without special exercise equipment, in any environment, and with high exercise effectiveness. Exercise mode adopts 30 s of exercise, 10 s of rest exercise cycle, performing jumping jacks, wall sits, push-ups, abdominal crunch, step-up onto a chair, squat, triceps dip on a chair, plank, high knees, lunge, push-up, and rotation, swap left and right—side plank, and 12 other dynamic and static exercise programs [47], which can achieve aerobic exercise effects to increase muscle endurance at the same time [48].

### 2.2. Massage

Massage, also known as tui na, is one of the oldest medical treatments in China [36]. It is a medical and health care method that uses the part of the hands for manipulation and was gradually developed by the ancient Chinese people after summarizing their knowledge and skills from long experiences of labor and disease care [37]. Massage can be divided into medical massage, health massage, and sports massage in terms of techniques, contents, and application objects [36,37]. According to research, massage has been shown to be useful in reducing cardiovascular and visual diseases and improving overall health quality [41].

Massage techniques are diverse and mainly aim at relieving fatigue and tension in patients [47]. Massage techniques and skills are diverse but are derived in similar ways from various branches. Common massage techniques include pushing, rubbing, kneading, kneading and pinching, pressing, scrubbing, tapping, shaking, pulling, and acupressure. They all aim to achieve the goal of patient comfort and effectiveness [49], and in fact, the effects are similar and do not differ much.

### 2.3. Seven-Minute High-Intensity Circuit Training (HICT) and Massage

HICT has positive effects on improving physical and mental health as well as physical fitness [27], achieving great results in a short time span. Therefore, the investigators believed that, under the pressure of the current epidemic environment, high-intensity intermittent exercise could resolve the barriers to exercise caused by time and space constraints and increase university students’ willingness to exercise.

Although exercise helps to improve health [22], long-term training produces periodic fatigue [12,44,45] and results in tiredness and muscle discomfort [16,29,30], which in turn affects the willingness to exercise. Studies have confirmed that massage helps to reduce exercise fatigue [36,37,38,39], prolong the ability to exercise, and preserve health. Therefore, it was hypothesized that massage should have the same fatigue-reducing effect on participants of intermittent exercise. However, since there have been no relevant research results, the investigators sought to investigate whether massage would have the same effect on the exercise performance and body composition of intermittent exercise participants.

### 2.4. Physical Fitness (Exercise Performance)

Healthy physical fitness is composed of four different physical abilities, including cardiorespiratory fitness, muscular fitness, flexibility, and body composition [50]. Physical fitness can be assessed in terms of cardiorespiratory fitness, muscular fitness, flexibility, and body composition [47].

Cardiopulmonary fitness is measured in seconds and is usually performed by running and walking. The smaller the number, the better the cardiorespiratory fitness [51]. It is used to understand the function of gas exchange in the lungs and the rate at which the heart transports blood and oxygen and other nutrients to the whole body to provide energy for overall physical activity [23].

The muscle fitness test is evaluated by one-minute bent knee sit-ups and standing long jump and measured in terms of the number of reps and centimeters, the larger the number, the better the muscle fitness [51]. It is used to understand the strength of the abdominal and lower limb muscle groups of the individual.

Flexibility is determined by measuring how far the body is bent forward in a seated position, measured in centimeters, and the longer the measurement distance, the better the result.

Body composition is represented by the body mass index (BMI), and the standard value falls between 18 and 25 (standards vary for men and women). The measurement formula is:BMI = weight (kg)/height (m^2^)

Therefore, to understand the effects of intermittent exercise on physical fitness and athletic performance of university students, corresponding test indicators and methods are needed. The physical fitness and exercise performance will be measured by 800/1500 running and walking, one-minute bent knee sit-ups, standing long jump, and seated forward bend, and the physiological health indicators will be assessed by body weight, height, blood pressure, and BMI.

### 2.5. Physical and Mental Health Awareness

As stated by WHO in 1948, physical and mental health refers to a state of physical, mental, and social well-being [52]. This includes the physical, psychological and social aspects of an individual’s body composition, mental health, and social adjustment [53].

Physical and mental health is an analytical method of self-perception assessment [54,55], and public health issues, moreover, require scientific evidence to present the actual results [56]. Investigating individual physical and mental health phenomena based on personal perceptions can reveal the impact of the current environment on people [20]. Physical and mental health can be viewed in terms of psychological, spiritual, and attitudinal components [20,57,58], as evidenced by anxiety, competence, enthusiasm, headache, abdominal pain, insomnia, stomach pain, irregular diet, and suicidal ideation [59,60].

Therefore, the researchers sought to understand whether the physical and mental health of university students who used massage to relieve muscle and psychological stress after intermittent exercise were relieved. The study of the psychological, mental, and attitudinal aspects of the subjects would provide accurate information. As shown in Figure 1.

## 3. Methods and Instruments

### 3.1. Study Framework, Design, Positioning, and Hypotheses

Exercise is beneficial for physical and mental health [22]. 7-min high-intensity circuit training can be used in a safe environment, without special exercise equipment, in any setting to obtain high exercise effectiveness [47]. Although intermittent exercise will produce sports fatigue, massage helps to recover from fatigue, effectively activates blood circulation, and improves body quality [41,61]. This exercise theory and model may be effective in improving the physical and mental health of university students. Therefore, based on the above theory, a mixed research method [62,63,64,65,66,67] was used to obtain 20 volunteers and divide them into an experimental and a control group. During the experimental period, both groups were tested to perform a 7-min intermittent exercise experiment, and their physical fitness performance, body composition, and body composition were monitored. The questionnaires were compiled with reference to the literature [24,25,26,27,28,29,30,31,32,33,34,35,36,37,38,39,40,41,42,43,44,45,46,47,48,49,50,51,52,53,54,55,56,57,58,59,60], and the subjects’ physical and mental health cognitions were continuously collected and validated by statistical validation using SPSS 26.0 statistical software. Lastly, all the information was gathered and integrated by a rigorous sequence of consolidation, organization, and analysis so as to finally construct the article [68] and analyze the results with multivariate verification [69,70].

Based on the above description, there are three research hypotheses:

**Hypothesis** **1**(**H1**)**.** *It is hypothesized that massage would result in a significant improvement in the physical fitness performance of intermittent exercise participants.*

**Hypothesis** **2**(**H2**)**.** *It is hypothesized that massage would result in a significant improvement in the body composition of intermittent exercise participants.*

**Hypothesis** **3**(**H3**)**.** *It is hypothesized that there would be significant improvements in physical and mental health for participants with or without sports massage combined with intermittent exercise.*

### 3.2. Study Procedure and Tools

The study was conducted to verify the effects of massage on the physical fitness performance, body composition, and physical and mental health of intermittent exercise participants. Two types of research designs were used: An ambient exercise experiment and a questionnaire survey. The research process and tools are described below.

#### 3.2.1. Experimental Design of Physical Fitness Testing

Due to the regulation and requirement of exercise planning and the fact that the exercise performance could not reach the intermittent exercise performance, the final number of subjects in the physical fitness test experiment was 20. In addition to the subjects’ daily routine, the experimental group adopted a 7-min high-intensity intermittent exercise design and performed a sports massage after the exercise; the control group did not receive a sports massage, and the two groups conducted the experiment for 7 weeks.

Tested on the day before and after the intermittent exercise training for 7 weeks. The experiment was conducted using a digital watch (timing), a multifunctional smartphone (video playback, recording, and audio recording), a leather ruler (measuring height and flexibility), an oximeter, and a bodyweight meter to measure the basic values of height, weight, blood pressure, blood oxygen concentration, and trimesters, and to record the 800/1500 m running time.

In the study design, the physical fitness and body composition data were measured before and after the experiment on the first and last day, and the participants’ body composition values such as height, weight, blood pressure (pulse), blood oxygen concentration, and trimesters were measured. The participants were also tested in 800/1500 m running and walking, standing long jump, seated forward bend, and knee-bending sit-ups.

On the first day, after obtaining data on physical fitness and body composition, the experiment was explained and massage knowledge and skills were reinforced. On the second day, according to the experimental program of 5 consecutive days per week for 7 weeks, body composition and blood pressure (pulse) data were obtained before each operation, and a 7-min intermittent exercise course was conducted afterward. A total of 12 exercise movements were performed, with 30 s of operation and 10 s of rest for each movement. After the exercise training was completed, the exercise massage was immediately carried out for 5–10 min before the exercise experiment program was completed.

#### 3.2.2. The Compiling and Analysis of Mental Health Questionnaire

The physical and mental health scale was developed with reference to relevant literature [23,24,25,26,27,28,29,30,31,32,33,34,35,36,37,38,39,40,41,42,43,44,45,46,47,48,49,50,51,52,53,54,55,56,57,58,59,60], and then three coaches with national sports team coaching experience and academics in the fields of exercise physiology and psychology were invited to review the contents of the questionnaire. Ten questionnaires were distributed initially, and the reliability and validity of the questionnaires were examined using SPSS 26.0 statistical software, and a high-reliability question with a Cronbach’s alpha of 0.7 or higher was used for follow-up analysis [71]. However, due to the environmental safety considerations of the COVID-19 epidemic and the specific exercise program and target population, the questionnaires were distributed and collected using the Chinese Wenjuanxing online questionnaire platform. Intentional sampling was used and the participants of this experiment were defined as the subjects. A total of 120 subjects were originally included in the survey, but due to elimination and withdrawal during the participation period, a total of 66 questionnaires were obtained including the participants. Basic statistics were used to examine the post-experimental self-awareness questionnaires of the participants. VOOV video conferencing software was used to interview the participants, experts, and scholars, and opinions were presented based on the statistical test results. Finally, the multivariate verification analysis method was used to integrate the information from different research subjects [69,70] by various research theories and methods, and accurate knowledge and meanings were obtained by comparing the research results from multiple perspectives and data [68].

Self-perception assessment of physical and mental health [54,55] examines psychological, spiritual, and attitudinal components [57,58] in terms of anxiety, competence, enthusiasm, headache, abdominal pain, insomnia, stomach pain, eating irregularities, and suicidal intention [59,60,62]. There were 13 questions on physical and mental health, and the results of statistical analysis showed that the KMO value was 0.671, while Bartlett’s approximate χ^2^ value was 185.86 and df was 78, with a significance of *p* < 0.001, which was suitable for factor analysis. The explained variances of the scales were 34.95%, 25.13%, and 13.88%, and the total explained variance was 73.97%. After factor analysis, all of them were retained. The alpha coefficients of the three scales were 0.917, 0.909, and 0.905, and the alpha coefficient of the total scale was 0.920. Based on the above analysis results, it is clear that this questionnaire has good reliability. As shown in Table 1.

### 3.3. Study Scope and Limitations

A mixed-method study was conducted to verify the effects of massage on physical fitness performance, body composition, as well as physical and mental health of intermittent exercisers using experimental, quantitative, and qualitative studies. The subjects of the study were students aged about 20 years old from a university in Fujian, China. Due to safety considerations such as schooling, daily life, personal physical health, and athletic performance ability, as well as factors such as funding, time, and manpower, it was not possible to provide a sample size for validation and analysis, so it was not possible to extend the validation to other countries, ages, or occupational backgrounds. The above limitations will be listed as suggestions for follow-up research, awaiting further studies by other researchers for improvement.

### 3.4. Ethical Considerations

The data for this study were collected by a combination of intentional sampling and snowball sampling, and the participants were all students enrolled in the course. Therefore, all respondents approved and understood the purpose of the study and agreed to cooperate in providing relevant data. All questionnaires and interviews were recorded and data were collected anonymously and knowingly.

## 4. Analysis of Results

It is known in the literature that exercise improves physical and mental health [22], although intermittent exercise tends to produce fatigue, which affects exercise performance and willingness to participate [35]. Massage can help to relieve fatigue [37], stabilize cardiovascular disease [41], and restore physical and mental health [38]. According to the study design, the physical fitness performance, body composition, and physical and mental health awareness of the subjects were analyzed.

The study consisted of two parts: an experimental study and a questionnaire survey. All subjects were recruited as volunteers who were clear about the purpose of the experiment during the study. Under the condition that their living and learning needs were not affected, the subjects agreed to cooperate with the research project process and were willing to provide sample data during the experiment in an anonymous and non-public manner, thus the experimental data were considered highly valid. However, the experimental data may still be affected due to the experimental targets and the consideration of subjects’ health and safety. We, therefore, recruited students aged about 20 years old from a university in Fujian, China as the target population. Sixty-six participants, 30 males and 36 females, were enrolled in the questionnaire survey. The experimental study was conducted with 20 valid samples (7 males and 13 females), and the experimental group (with massage, 5 males and 5 females) and the control group (without massage, 2 males and 8 females) were assigned by intentional sampling.

### 4.1. Analysis of Physical Fitness Differences

The data of the experimental subjects were obtained from the experimental group (Tester) of 10 individuals, numbered T1–T10, and the control group (Controller) of 10 individuals, numbered C1–10. According to the physical fitness test standard [72], the performance of sit-ups, seated forward bends, standing long jump, and cardiorespiratory endurance were analyzed, and the assessment levels included gold, silver, bronze, medium, and to be enhanced.

The standard values for male and female testing in the assessment were:

The standard number of sit-ups within 60 s was gold (>44:>35), silver (42–43:33–34), bronze (38–41:28–32), medium (33–37:24–28), and to be enhanced (26–32:18–23).

The standard times of seated forward bend within 60 s are: gold (>41:>44), silver (38–40:41–43), bronze (32–37:35–40), medium (26–31:29–34), and to be strengthened (17–25:20–27).

The distance of the long jump was measured in centimeters during the two jump tests, and the longest distance was measured, gold (>235:>182), silver (245–249:175–178), bronze (229–241:162–172), intermediate (214–226:149–159), and to be strengthened (191–210:130–146).

Cardiorespiratory endurance was measured in seconds for single exercise performance in running (male: female), gold medal (<7′10:<4′10), silver medal (7′37–7′25:4′24–4′18), bronze (8′28–7′49:4′49–4′30), medium (9′18–8′37:5′10–4′54), and to be enhanced (10′30–9′31:5′52–5′21).

The analysis revealed that the subjects’ physical fitness changed after the experiment. The improvement in sit-ups ranged from a low of 1 to a high of 34, with women (+34) showing a greater improvement than men (+10). The improvement in seated forward bend ranged from a minimum of 0 cm to a maximum of 20 cm, with females (+20) improving more than males (+5). All subjects improved in the standing long jump, except for one female (−3) whose performance decreased, with a minimum improvement of 3 cm and a maximum improvement of 48 cm, and males improved more than females. All subjects improved in cardiorespiratory endurance, with a minimum improvement of 5 s and a maximum improvement of 1 min and 43 s (103 s), and the performance of females (1 min and 43 s) was more obvious than that of males (60 s). As shown in Table 2.

Before the experiment, all subjects in the control group performed 23–43 sit-ups, 0–60 cm seated forward bend, 138–220 cm standing long jump, 7′43 for men (1500 m), and 5′40 for women (800 m). After the experiment, all subjects in the control group showed growth in sit-ups, with a minimum of 2 and a maximum of 8, with females (+8) showing greater growth than males (+2). In the Seated forward bend event, all subjects showed growth, with a minimum of 1 cm and a maximum of 22 cm, and greater growth for females (+22) than males (+2). In the standing long jump event, except for two female (−5) subjects who decreased, all the subjects grew, with minimum growth of 5 cm and maximum growth of 13 cm, and the growth of male (+13) was higher than that of female (+10). In the cardiorespiratory endurance category, all subjects showed an increase in speed, with a minimum increase of 24 s and a maximum increase of 1 min and 10 s (70 s), and the improvement was more pronounced. As shown in Table 3.

Based on the above results, the experimental group was compared with the control group in sit-ups, seated forward bends, standing long jump, and cardiorespiratory endurance. It was found that the experimental group had better results than the control group in all items except for the seated forward bend (20:22). In both male and female subjects, the control group improved more than the experimental group in seated forward bend and cardiorespiratory endurance, while the experimental group outperformed the control group in sit-ups and cardiorespiratory endurance. This result indicates that Hypothesis 1 holds. As shown in Table 4.

Inspector: “When you are engaged in the seven-minute intermittent exercise, how does massage change the test results of your front and back exercise adaptability?”

T1: “Although intermittent exercise is intensive and frequent, it feels very effective …, massage can make me feel less fatigued during exercise.”

C1: “I feel tired easily during exercise, especially my thighs.”

T2: “Exercising makes me feel stronger and stronger in my thighs …; I will not affect my sports performance due to fatigue.”

C6: “In the initial stage, when the test was to be carried out, I always felt that I could not complete the test …, but after exercising, I did not expect to be able to complete the test. This means that whenever I feel fatigued after exercising, I will not be able to complete the test tomorrow. Will not be able to boost motivation.”

### 4.2. Analysis of Body Composition Differences

According to the test standard [73], BMI was calculated by weight and height data of individuals, and converted by the formula, and divided into underweight (18.5–), healthy (18.5–24), overweight (24–27), mild obesity (27–30), moderate obesity (30–35), and obesity (35+). The blood pressure was measured in mmHg, and the diastolic blood pressure was divided into normal (<120), initial (120–139), phase I (140–159), and phase II (>160), and the systolic blood pressure was divided into normal (<80), initial (80–89), phase I (90–99), and phase II (>100).

The analysis revealed that the experimental group had a height of 153–176 cm, a weight of 50.6–66.5 kg, a diastolic blood pressure of 64–100, a systolic blood pressure of 111–163, and a BMI of 21.4–17.9. After the experiment, the experimental group had no change in height, but had a minimum change of −0.2 and a maximum change of +1.22 in weight, with a greater change in men (−2.3) than in women (−2). The minimum change in BMI was −0.7 and the maximum change was −1.2, with a greater change in males (−1.4) than females (+0.3). In blood pressure, all subjects showed improvement. Diastolic blood pressure improved at a minimum of −4 and a maximum of −25, with a greater change in women (−25) than in men (−19); systolic blood pressure improved at a minimum of 0 and a maximum of −47, with a greater change in men (−47) than in women (−20). As shown in Table 5.

The analysis revealed that the control group initially had a height of 157–168 cm, a weight of 44.6–60 kg, a diastolic blood pressure of 68–80, a systolic blood pressure of 100–134, and a BMI of 16.5–23.7. After the experiment, there was no change in height in the control group, but the weight change was −2 at the minimum and +1.5 at the maximum, with a greater change in males (−2.3) than in females (−2.0). 

The change in BMI was 0 at the minimum and −0.7 at the maximum, with a greater change in males (−1.4) than in females (+0.3). Blood pressure improved in all subjects, with diastolic blood pressure improving at a minimum of +2 and a maximum of −1.4, with a greater change in women (+9) than in men (+5); systolic blood pressure improved at a minimum of 0 and a maximum of −19, with a greater change in women (+19) than in men (−3). As shown in Table 6.

According to the above results, there was no difference between the experimental group and the control group in terms of height measurements. In terms of body weight, the control group grew more than the experimental group (1.5 > 0.22). In terms of diastolic blood pressure, the experimental group improved more than the control group (−25 > −15) and systolic blood pressure (−47 > −19). The improvement in BMI was greater in the experimental group than in the control group (−1.2 > −0.7), and the improvement was greater in the experimental group than the control group for men and in the control group than the experimental group for women. This result indicates that Hypothesis 2 holds. As shown in Table 7.

Inspector: “When you are engaged in the seven-minute intermittent exercise, how does massage change your blood pressure and body composition (height, weight)?”

T2: “I feel more massaged, which reduces my fatigue problem, …, sleeps better, and usually feels calmer …”

C3: “Although I feel tired easily during exercise, I feel better breathing after exercise and when I get up every morning …; my appetite feels better …”

T10: “After more massages, exercise makes me feel better and better. Adding massage after exercise makes me feel better. When I get up the next day, I feel more energetic and emotionally stable …”

C1: “As the number of days of exercise increases, the fatigue will become more and more obvious …”

C4: “After each exercise, fatigue still exists, and it is becoming more and more obvious …; After a day of study and exercise, after a day of rest, you will feel more energetic.”

### 4.3. Physical and Mental Health Awareness Analysis

Next, the analysis of physical and mental health awareness revealed that all subjects believed that intermittent exercise could increase confidence (4.02), improve sleep (3.45), and increase appetite (2.63), but had little effect on relieving headaches (2.49), emotional disorders (2.77), and fear of depression (2.97). In addition, both the experimental and control groups showed significant (*p* > 0.01) differences in back pain and suicidal tendency, and the experimental group was better than the control group. There was no significant difference between the perceptions of men and women in the different experimental groups (*p* > 0.05). This result indicates that Hypothesis 3 is not valid. As shown in Table 8.

Inspector: “When you are engaged in the seven-minute intermittent exercise, how does massage change your physical and mental health before and after?”

T2: “I feel more massaged, which reduces my fatigue problem … After exercise, it makes me sleep better and my body feels calmer …; I feel that all the stress is gone.”

T4: “After exercising, I feel the pressure disappeared all day; my stomach muscles feel tighter and tighter.”

T5: “After each exercise, the waist muscles feel tight.”

C3: “Although exercise makes me feel more comfortable, the fatigue after exercise still makes me feel uncomfortable …”

T10: “After more massages, exercise makes me feel better and better. Adding massage after exercise makes me feel better. When I wake up the next day, I feel better …”

C1: “As the number of days of exercise increases, the feeling of fatigue will become more and more obvious …; The fatigue that cannot be eliminated, coupled with the pressure of study, will make you feel a little tired.”

C4: “If I can, I hope that there will be no fatigue after exercise. Soreness will cause problems in my studies.”

### 4.4. Discuss

#### 4.4.1. Physical Fitness Difference Analysis

The main physical fitness tests are sit-ups, seated forward bends, standing long jump, and cardiorespiratory endurance. Sit-ups and cardiorespiratory endurance require a high level of muscular endurance performance. The standing long jump requires sufficient explosive power and body coordination. Seated forward bend requires flexibility of the limbs and comfortable muscle and joint extension to achieve good athletic performance.

The experiments confirmed that the intermittent exercise participants who received the massage (experimental group) had better results than the group without the massage (control group) in all physical fitness measures, except for the seated forward bend.

We believed that although intermittent exercise features short time requirements and high efficiency [29], long-term training results in periodic fatigue [16,47,48], which can be a nuisance for exercise participants. Massage has a significant effect on relieving fatigue [49] and improving physical fitness [41]. For the experimental group, although fatigue might occur, the massage technique could be used to accelerate the metabolism of lactic acid, eliminate fatigue, and restore muscle comfort. While the training objectives of the exercise program in this study were mainly about muscular fitness and cardiorespiratory endurance training and not about physical extension activities, the participants still benefited from massage techniques that restore their muscle flexibility and eliminate fatigue. Therefore, in addition to strengthening the existing exercise objectives, enhancing the endurance and function of muscles and the cardiorespiratory system can also slightly improve muscular extensibility.

Consequently, the experimental group had better results than the control group in all tests except for the seated forward bend. In both male and female subjects, the control group improved more than the experimental group in the seated forward bend and cardiorespiratory endurance tests, while the experimental group outperformed the control group in sit-ups and cardiorespiratory endurance.

#### 4.4.2. Body Composition Difference Analysis

The results showed that although the bodyweight of the experimental group increased, the improvement in blood pressure (diastolic and systolic) and BMI was greater than that of the control group.

We believed that massage helps to deter cardiovascular disease [41] and intermittent exercise helps to improve physiological status [33]. The integration of exercise planning and massage can help to obtain better exercise results. In this study, the main objective of the training was to improve cardiopulmonary fitness and muscular endurance. Good cardiopulmonary fitness helps to increase oxygen intake and improve blood circulation while relieving psychological stress, which can improve physical fitness to a certain extent. However, most of the subjects had already passed the growth period, and the experiment was only conducted for a short time span. If we want to extend the investigation of the effect of intermittent exercise on growth, it will be necessary to extend the experiment period or find other subjects for the experiment.

In addition, in the early stage of the experiment, the BMI of the participants were generally between 16.5–23.7 which indicated a healthy or thin state (healthy: 18.5–24). Because exercise stimulates blood circulation and accelerates metabolism, the body needs to generate a considerable amount of energy to cope with the high-intensity exercise. In order to obtain sufficient energy, the participants thus increased their nutritional intake, which in turn changed their health status. However, excessive dietary intake may lead to nutrient accumulation and fatigue from exercise, which may affect exercise performance and willingness to burn calories, resulting in insufficient exercise performance. Therefore, the results of the study showed that there was no difference in the height measurement for all subjects. For blood pressure and BMI, the experimental group improved better than the control group, and the control group gained more weight than the experimental group.

#### 4.4.3. Physical and Mental Health Awareness Difference Analysis

The results showed that all intermittent exercise participants significantly felt a boost in confidence, improved sleep, and increased appetite, but there was no improvement in headache, loss of emotional control, or fear of depression. The data indicated that the control group experienced more pronounced back pain and suicidal tendencies, and there was no difference in the feelings of the different genders in each group.

We believed that due to the improved life quality and excessive nutritional intake [2], coupled with environmental risks in the community [8] that change the lifestyle and willingness to exercise, youths are at risk for physical and mental health problems [9]. Exercise is beneficial to physical and mental health [22]. High-intensity intermittent exercise can further increase metabolism, relieve stress, and effectively improve physical and mental health. Due to its short duration and fast exercise effect, high-intensity intermittent exercise has become an ideal option for young people to improve their physical and mental health. In addition, massage can eliminate the fatigue generated after exercise, promote blood circulation, accelerate metabolism, improve muscle tension, and relieve stress. However, because of the busy schedule of university students and the heavy load of homework, coupled with the high intensity of intermittent exercise, those who are not participants in long-term sports or have no exercise planning may not be able to relieve the pressure of engaging in exercise and still feel nervous, fear, and even have headache problems. Therefore, all subjects thought that intermittent exercise could increase confidence, improve sleep, and increase appetite, but it was not very effective in relieving the feelings of headache, loss of emotional control, and fear of depression.

Although intermittent exercise is highly intense and demanding, massage can be performed after exercise to relieve muscle pain. The exercise carried out in this study mostly aimed to improve cardiorespiratory endurance, which can effectively promote blood circulation, enhance concentration and sports performance, and further increase self-confidence. However, because they were mostly endurance training and abdominal muscle exercises, if the massage was not correctly performed or insufficient, it might not be able to relieve the fatigue of deep abdominal muscles, resulting in the problem of back pain. Therefore, although the experimental group was prone to the problem of back pain due to exercise, it was effective in improving self-confidence and reducing suicidal ideation.

## 5. Conclusions

The findings of the study showed that intermittent exercise for university students can help enhance physical fitness and sports performance, improve body composition, and regulate physical and mental health status. If combined with sports massage, physiologically, it can improve the performance of sit-ups, standing long jump, blood pressure, and BMI. Psychologically, it can improve self-confidence and reduce the negative thoughts of seeking death. However, sports massage has little effect on the negative psychological feelings of headache, emotional loss of control, fear or depression, and physiological phenomena such as flexibility and cardiorespiratory endurance of participants of intermittent exercise.

Based on the findings of the study, the following recommendations are made:1.Establishing intermittent exercise programs for university students to improve their physical and mental health.

According to the results of the experiment, intermittent exercise is beneficial to improve physical and mental health, and schools should be encouraged to introduce relevant exercise courses to help university students establish good exercise habits in the face of a reduced social and sports environment.

2.Adjusting intermittent exercise content to meet different needs.

Intermittent exercises have certain restrictions and intensities, which cannot be easily adapted due to the physical and mental health of college students. Therefore, it is recommended that participants should adjust the intensity of exercises on their own based on their health conditions, or design exercise programs that train different body components, muscles, or tissues to provide participants with choices.

3.Continuing exploration in light of the study limitations and findings.

The study found that massage significantly affected the performance of intermittent exercise participants. However, due to the limitations of the study, the changes in performance during exercise were not explored in depth. In addition, this study only recruited Chinese university students as the experimental subjects, so the findings may not be applicable to other countries or age groups of exercise participants. Therefore, it is recommended that subsequent researchers extend the study to examine changes in exercise performance or explore other regions and subjects to help fill in the research gaps.

## Figures and Tables

**Figure 1 ijerph-18-05013-f001:**
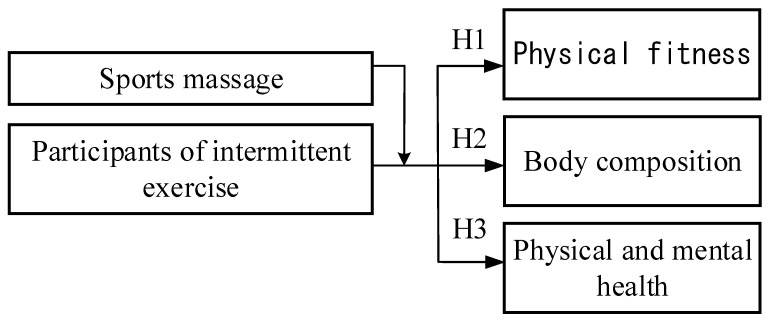
The research structure.

**Table 1 ijerph-18-05013-t001:** Analysis of questionnaire tools.

Construct	Dimension	
Basic Variables	Gender (Male/Female), Identity (Experimental Participant/Controller)	
Construct	Dimension	Cronbach’s α
Psychological feelings	01. Make me feel good and increase my confidence in facing things around me	0.917
	02. It makes me feel scared and scared	0.912
	03. Satisfied with my work performance	0.915
	04. Let me be passionate about things or activities	0.912
	05. Make me feel that I can use my time to do things	0.916
Mental status	06. Makes me feel headache or pressure on the head	0.909
	07. It makes me feel backache	0.908
	08. Make me sleepless or sleep well	0.907
Life attitude and health	09. It makes me feel stomachache and indigestion	0.905
	10. Let me have an abnormal diet (eat more, drink more, or smoke more)	0.902
	11. Makes me impatient and easy to lose my temper	0.903
	12. I still feel that work and life are meaningless and I feel very lost	0.905
	13. I still feel that death can escape everything	0.900

**Table 2 ijerph-18-05013-t002:** Statistical difference analysis of physical fitness of the experimental group before and after the experiment.

	**T1 (Male)**			**T2 (Male)**			**T3 (Male)**			**T4 (Male)**		
	**Before**	**After**	**DIFF**	**Before**	**After**	**DIFF**	**Before**	**After**	**DIFF**	**Before**	**After**	**DIFF**
A	30	40	+10	15	17	+2	30	41	+11	15	18	+3
B	10	12	+2	30	35	+5	10	13	+3	30	38	+3
C	180	208	+48	210	212	+2	180	207	+27	210	215	+5
D	7′35	6′33	−′60	7′05	6′15	−′50	7′35	6′30	−′60	7′05	6′12	−′53
	**T5 (Male)**			**T6 (Female)**			**T7 (Female)**			**T8 (Female)**		
	**Before**	**After**	**DIFF**	**Before**	**After**	**DIFF**	**Before**	**After**	**DIFF**	**Before**	**After**	**DIFF**
A	50	51	+1	30	36	+6	46	52	+6	30	36	+6
B	12	12	+0	50	53.5	+3.5	30	36	+6	13	36	+20
C	200	236	+36	160	164	+4	165	168	+3	150	155	+5
D	7′30	6′43	−′37	5′30	4′30	−′60	4′39	3′52	−′47	5′23	3′50	−1′43
	**T9 (Female)**			**T10 (Female)**								
	**Before**	**After**	**DIFF**	**Before**	**After**	**DIFF**						
A	29	35	+6	29	63	+34						
B	93	97	+4	61	63.5	+2.5						
C	163	160	−3	150	163	+13						
D	5′01	4′13	−′58	4′30	4′35	−′5						

A, sit-up; B, seated forward bend; C, standing long jump; D, cardiorespiratory endurance.

**Table 3 ijerph-18-05013-t003:** Statistical difference analysis of physical fitness of the control group before and after the experiment.

	**C1 (Male)**			**C2 (Male)**			**C3 (Female)**			**C4 (Female)**		
	**Before**	**After**	**DIFF**	**Before**	**After**	**DIFF**	**Before**	**After**	**DIFF**	**Before**	**After**	**DIFF**
A	43	45	+2	40	42	+2	42	47	+4	31	35	+4
B	0	12	+12	3	16	+13	60	61	+1	55	65	+10
C	220	230	+10	202	215	+13	175	177	+2	160	160	+0
D	7′43	6′42	−′59	7′58	6′55	−′58	4′16	3′54	−′24	4′53	3′50	−1′03
	**C5 (Female)**			**C6 (Female)**			**C7 (Female)**			**C8 (Female)**		
	**Before**	**After**	**DIFF**	**Before**	**After**	**DIFF**	**Before**	**After**	**DIFF**	**Before**	**After**	**DIFF**
A	40	43	+3	30	32	+2	28	32	+4	41	45	+4
B	45	47	+2	40	42	+2	40	45	+5	25	28	+3
C	160	170	+10	150	158	+8	138	145	+7	170	175	+5
D	4′54	4′20	−′34	5′30	5′15	−′15	5′40	5′35	−′5	4′30	4′01	−′29
	**C9 (Female)**			**C10 (Female)**								
	**Before**	**After**	**DIFF**	**Before**	**After**	**DIFF**						
A	23	26	+3	32	40	+8						
B	13	25	+12	13	35	+22						
C	165	160	−5	165	160	−5						
D	4′30	3′50	−′40	5′00	3′50	−1′10						

A, sit-up; B, seated forward bend; C, standing long jump; D, cardiorespiratory endurance.

**Table 4 ijerph-18-05013-t004:** Comparative analysis of exercise performance data between experimental group and control group.

	Test				Control			
	Lowest	Highest	Male	Female	Lowest	Highest	Male	Female
A	1	34	10	+34	2	8	2	+8
B	0	20	5	+20	1	22	2	+22
C	3	48	48	±3	±5	13	13	+10
D	5	103	60	+103	24	70	70	+59

A, sit-up; B, seated forward bend; C, standing long jump; D, cardiorespiratory endurance.

**Table 5 ijerph-18-05013-t005:** Statistical comparison of body quality and physical fitness of subjects in the experimental group before and after the experiment.

			**T1 (Male)**			**T2 (Male)**			**T3 (Male)**			**T4 (Male)**		
			**Before**	**After**	**DIFF**	**Before**	**After**	**DIFF**	**Before**	**After**	**DIFF**	**Before**	**After**	**DIFF**
cm			176	176	+0	173	173	+0	176	176	+0	173	173	+0
kg			62.2	63.4	+1.2	62.3	61.1	−1.2	62.3	64.6	−2.3	62.3	61.3	−1
Kpa		D	84	76	−8	100	81	−19	88	78	−10	90	80	−10
		S	136	120	−16	163	116	−47	137	122	−15	140	115	−25
BMI			20	20.4	+0.4	18	17.7	−0.3	20	20.8	+0.8	18	17.7	−0.3
			**T5 (Male)**			**T6 (Female)**			**T7 (Female)**			**T8 (Female)**		
			**Before**	**After**	**DIFF**	**Before**	**After**	**DIFF**	**Before**	**After**	**DIFF**	**Before**	**After**	**DIFF**
cm			173	173	+0	163	163	+0	163	163	+0	168	168	+0
kg			66.5	64.7	−1.8	52.5	53	−0.5	50	50.6	−0.6	58	56	−2
Kpa	D		90	79	−11	73	80	−7	80	63	−17	74	78	−4
	S		128	118	−10	103	111	−8	117	117	+0	117	108	−9
BMI			20.2	21.6	+1.4	19.7	19.4	+0.3	15.3	15.5	−0.2	20.5	19.8	−0.7
			**T9 (Female)**			**T10 (Female)**								
			**Before**	**After**	**DIFF**	**Before**	**After**	**DIFF**						
cm			173	173	+0	153	153	+0						
kg			53.7	53	−0.7	50.5	50.3	−0.2						
Kpa		D	67	73	+6	89	64	−25						
		S	114	108	−6	111	90	−20						
BMI			17.9	17.7	−0.2	21.4	21.4	+0						

D (diastolic blood pressure), S (systolic blood pressure).

**Table 6 ijerph-18-05013-t006:** Statistical comparison of body quality and physical fitness of subjects in the control group before and after the experiment.

			**C1 (Male)**			**C2 (Male)**			**C3 (Female)**			**C4 (Female)**		
			**Before**	**After**	**DIFF**	**Before**	**After**	**DIFF**	**Before**	**After**	**DIFF**	**Before**	**After**	**DIFF**
cm			165	165	+0	166	166	+0	157	157	+0	163	163	+0
kg			49.5	51	+1.5	50.5	51	−0.5	46.5	45.8	+0.7	60	58.7	+1.3
Kpa		D	70	75	+5	79	74	+5	64	65	−1	80	71	+9
		S	115	112	−3	118	115	+3	108	102	+6	134	115	−19
BMI			18.1	18.4	−0.3	18.3	18.7	−0.4	18.6	18.5	+0.1	22.5	22	+0.5
			**C5 (Female)**			**C6 (Female)**			**C7 (Female)**			**C8 (Female)**		
			**Before**	**After**	**DIFF**	**Before**	**After**	**DIFF**	**Before**	**After**	**DIFF**	**Before**	**After**	**DIFF**
cm			165	165	+0	157	157	−29.5	157.5	157	+0.5	168	168	+0
kg			45	47	−2	51.4	51.1	+0.3	51.4	51.1	+0.3	49	50	−1
Kpa	D		68	66	+2	72	69	+3	72	69	+3	74	65	+9
	S		102	118	−16	114	107	+7	114	107	+7	112	105	+7
BMI			16.5	17.2	−0.7	20.7	20.7	+0	20.7	20.7	+0	18.4	18.8	+0.4
			**C9 (Female)**			**C10 (Female)**								
			**Before**	**After**	**DIFF**	**Before**	**After**	**DIFF**						
cm			157	157	0	159	159	0						
kg			44.6	44.5	+0.1	58.5	58	+0.5						
Kpa		D	71	86	−15	81	78	3						
		S	100	100	0	114	115	−1						
BMI			18.1	18	+0.1	23.1	22.9	+0.2						

D (diastolic blood pressure), S (systolic blood pressure).

**Table 7 ijerph-18-05013-t007:** Comparison of body composition data between experimental and control groups.

		Test				Control			
		Lowest	Highest	Male	Female	Lowest	Highest	Male	Female
cm		0	0	0	0	0	0	0	0
kg		−0.2	+1.22	−2.3	−2	−2	+1.5	−2.3	−2
Kpa	D	−4	−25	−19	−25	+2	−25	+5	−15
	S	0	−47	−47	−20	0	−20	−3	−19
BMI		−0.7	−1.2	−1.4	0.3	0	−0.7	−0.4	+0.7

D (diastolic blood pressure), S (systolic blood pressure).

**Table 8 ijerph-18-05013-t008:** Analysis of physical and mental health awareness of subjects with different experimental contents.

		M	Identity (Tester/Controller)	Gender Test-Male/Test-Female/Control-Male/Control-Female
Psychological feelings	Make me feel good and increase my confidence in facing things around me	4.02	(4.17:3.99)	(4.25: 4.14:3.86:4.04)
	It makes me feel scared and scared	2.97	(3.99:3.15)	(2.50: 2.21:3.32:3.07)
	Satisfied with my work performance	3.91	(2.28:3.91)	(4.00:3.86:4.00:3.87)
	Let me be passionate about things or activities	3.90	(3.15:3.90)	(4.00:3.86:3.91:3.89)
	Make me feel that I can use my time to do things	3.79	(3.89:3.87)	(3.50:3.50:3.77:3.91)
Mental status	Makes me feel headache or pressure on the head	2.77	(3.91:2.93)	(2.50:2.07:3.05:2.87)
	It makes me feel backache	3.09	(3.89:3.22) *	(2.75:2.57:3.27:3.20)
	Make me sleepless or sleep well	3.45	(3.90:3.53)	(3.75:3.00:3.86:3.37)
Life attitude and health	It makes me feel stomachache and indigestion	2.60	(3.50:2.69)	(2.50:2.21:2.77:2.65)
	Let me have an abnormal diet (eat more, drink more, or smoke more)	2.63	(3.87:2.71)	(2.00:2.43:2.91:2.61)
	Makes me impatient and easy to lose my temper	2.49	(2.17:2.59)	(2.50:2.00:2.73:2.52)
	I still feel that work and life are meaningless and I feel very lost	2.38	(2.93:2.50)	(2.25:1.86:2.82:2.35)
	I still feel that death can escape everything	2.15	(2.61:2.28) *	(2.00:1.57:2.64:2.11)

* *p* < 0.05.

## Data Availability

No data support.

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
