# Peer review of "Effects of Sports Massage on the Physiological and Mental Health of College Students Participating in a 7-Week Intermittent Exercises Program"

_ijerph, 2021, doi:10.3390/ijerph18095013_

Round 1

Reviewer 1 Report

This is a very interesting paper about the effect of sports massage on exercise plan of college students. It is a good idea, but the paper must be improved.

I'll try to help you

1) pay attention to graphic: what are those green marks on keywords?

2) "A mixed-method approach was used to analyze the current status of athletic performance and body com-position of the volunteers through an experimental study, fol-lowed by the design of a questionnaire to interview the partici-pants on their physical and mental health, followed by inter-views with experts and scholars to provide opinions on the study results, and finally, a multivariate verification was conducted" --> what does it mean? what are methods of this paper? please state it clearly in the abstract section: this is one of the most important part of the article!

3) pay attention to grammar and to english style all over the paper!
"The study found that massage did have a significant effect" --> blue pen error!

4) I suggest to improve your references:
page 2 - "the risk of young people is increasing" --> https://pubmed.ncbi.nlm.nih.gov/31802420/
page 2 - "and sedentary or ly-ing-down time is increasing" --> https://pubmed.ncbi.nlm.nih.gov/32512767/ 
page 2 - "more than 70% of adolescents worldwide have low activity levels and inadequate physical adaptability" --> https://www.ncbi.nlm.nih.gov/pmc/articles/PMC4953112/ 
page 2 - "regulating psychological stress" --> https://www.mdpi.com/2411-5142/5/2/33 

5) please state in a more coincise way the aim of your study at the end of the introduction

6) page 6: "Finally, all the information
was compiled, and a rigorous sequence of compilation,
organization, and analysis was used to c" --> c what?

7) at the end of page 8 there is a different microsoft word style of the paragraph

8) page 9: Results, not Analysis of the results. Please write this section in a more simple way..too many numbers

9) Conclusion section: "Intermittent exercise helps college students to improve physical fitness, sports performance, body composition, as well as physical and mental health, and the combination of intermit-tent exercise and sports massage can again improve the perfor-mance of intermittent exercise participants in sit-ups, standing long jump, etc., improve blood pressure, affect BMI, improve self-confidence, and reduce suicide attempts." --> too long phrase! cut it into little ones

Author Response

Review 1

Dear reviewer

Thank you for your suggestion.

The content in blue font below is the description of the correction or reply suggestion.

1) pay attention to graphic: what are those green marks on keywords?

Dear reviewer

We have corrected the other color markings on that graph.

2) "A mixed-method approach was used to analyze the current status of athletic performance and body com-position of the volunteers through an experimental study, fol-lowed by the design of a questionnaire to interview the partici-pants on their physical and mental health, followed by inter-views with experts and scholars to provide opinions on the study results, and finally, a multivariate verification was conducted" --> what does it mean? what are methods of this paper? please state it clearly in the abstract section: this is one of the most important part of the article!

Dear reviewer

We have added the content.

3) pay attention to grammar and to english style all over the paper!
"The study found that massage did have a significant effect" --> blue pen error!

Dear reviewer

We have made drastic corrections.

4) I suggest to improve your references:
page 2 - "the risk of young people is increasing" --> https://pubmed.ncbi.nlm.nih.gov/31802420/
page 2 - "and sedentary or ly-ing-down time is increasing" --> https://pubmed.ncbi.nlm.nih.gov/32512767/ 
page 2 - "more than 70% of adolescents worldwide have low activity levels and inadequate physical adaptability" --> https://www.ncbi.nlm.nih.gov/pmc/articles/PMC4953112/ 
page 2 - "regulating psychological stress" --> https://www.mdpi.com/2411-5142/5/2/33 

Dear reviewer

Thank you for your information, we have added the content in the article.

5) please state in a more coincise way the aim of your study at the end of the introduction

Dear reviewer

We have made corrections.

6) page 6: "Finally, all the information
was compiled, and a rigorous sequence of compilation,
organization, and analysis was used to c" --> c what?

Dear reviewer

We have made corrections.

7) at the end of page 8 there is a different microsoft word style of the paragraph

Dear reviewer

We have made corrections.

8) page 9: Results, not Analysis of the results. Please write this section in a more simple way..too many numbers

Dear reviewer

We have made corrections.

9) Conclusion section: "Intermittent exercise helps college students to improve physical fitness, sports performance, body composition, as well as physical and mental health, and the combination of intermit-tent exercise and sports massage can again improve the perfor-mance of intermittent exercise participants in sit-ups, standing long jump, etc., improve blood pressure, affect BMI, improve self-confidence, and reduce suicide attempts." --> too long phrase! cut it into little ones.

Dear reviewer

We have made corrections.

The blue font is for corrections or supplements.

Thank you for the information, we have made a drastic correction. We believe in making articles more visual.

Reviewer 2 Report

I congratulate the authors for conducting such an interesting study. The present study aimed to examine the physical and mental health of college students after intermittent exercise with massage. Besides, numerous problems regarding the research were observed. For example, it seems that the amount of explanations within the text is unbalanced. 

  1. The manuscript stated as a case report, but it is a research study.
  2. The manuscript was written in sloppy English, making it hard to comprehend. Suggest sending the manuscript for English proofreading.
  3. The title and abstract also different; the title using interval, but the abstract is intermittent exercise. It is rather confusing. 
  4. The introduction is also sloppy; why COVID-19 was mention with the relationship to intermittent training are loosely explain.
  5. What type of massage? Chinese massage? What is the method of massage? 
  6. The introduction needs a stronger justification for the purpose of this study.
  7. Then, suddenly, high-intensity circuit training (HICT) was mention? 
  8. The arrangement of literature needs more attention – please rearrange it accordingly. 
  9. This study requires human ethical approval. If not, please state how the participants' vulnerability was ensured, as college students were recruited? 
  10. Please state the hypothesis for this study; otherwise, authors cannot interpret the p-value as significant or not significant.
  11. Where is the sample size calculation for the study? 
  12. Please change SPSS 22.0 software, it is long obsolete, and currently, only SPSS 25 or 26.0 are used. 
  13. Move the conceptual framework (Figure 1) to the literature review before methods. 
  14. The problem is motivation and blinding were not there; thus, participants who received additional “treatment – in this case, massage” will perform better. 
  15. What questionnaire was used to measure mental health? Is the questionnaire valid and reliable? It was not mentioned. 
  16. What is the analysis used? Not mentioned in a section. 
  17. To analyse, suggest using mixed-factorial ANOVA to analyse the results and not report individual data. 
  18. Unfortunately, the discussion was not based on the results. Please add more linkage from the finding. 
  19. I felt the study need stronger justification, conceptual framework etc. By missing so much information, it creates more doubts for the reader than answers. 
  20. References – please follow MDPI format. 

Unfortunately, I feel that the current manuscript was not sufficient to the required level for publication. But, if the authors could amend the suggested comment, it has a good chance. All the best.  

Thank you. 

Author Response

Review 2

Dear reviewer

Thank you for your suggestion.

The content in blue font below is the description of the correction or reply suggestion.

  1. The manuscript stated as a case report, but it is a research study.

Thank you reviewer

We agree with your views.

  1. The manuscript was written in sloppy English, making it hard to comprehend. Suggest sending the manuscript for English proofreading.

Thank you reviewer

We have made substantial corrections.

  1. The title and abstract also different; the title using interval, but the abstract is intermittent exercise. It is rather confusing.

Thank you reviewer

We have modified it to "Effects of Sports Massage on the Physiological and Mental Health of College Students' 7-week Intermittent Cycling Exercises Plan".

  1. The introduction is also sloppy; why COVID-19 was mention with the relationship to intermittent training are loosely explain.

Dear reviewer

We have made corrections.

  1. What type of massage? Chinese massage? What is the method of massage?

Dear reviewer

We have added the content in the article.

  1. The introduction needs a stronger justification for the purpose of this study.

Dear reviewer

We have made drastic corrections.

  1. Then, suddenly, high-intensity circuit training (HICT) was mention?

Dear reviewer

We have made drastic corrections.

  1. The arrangement of literature needs more attention – please rearrange it accordingly.

Dear reviewer

We have made drastic corrections.

  1. This study requires human ethical approval. If not, please state how the participants' vulnerability was ensured, as college students were recruited?

Dear reviewer

We have made drastic corrections.

  1. Please state the hypothesis for this study; otherwise, authors cannot interpret the p-value as significant or not significant.

Dear reviewer

We have made drastic corrections.

  1. Where is the sample size calculation for the study?

Dear reviewer

We have made drastic corrections.

  1. Please change SPSS 22.0 software, it is long obsolete, and currently, only SPSS 25 or 26.0 are used. 

Dear reviewer

We have made drastic corrections.

Amended to SPSS 26.0

  1. Move the conceptual framework (Figure 1) to the literature review before methods.

Dear reviewer

We have made drastic corrections.

  1. The problem is motivation and blinding were not there; thus, participants who received additional “treatment – in this case, massage” will perform better. 

Dear reviewer

We have made drastic corrections.

  1. What questionnaire was used to measure mental health? Is the questionnaire valid and reliable? It was not mentioned. 

Dear reviewer

We have made drastic corrections, and we have added the content in the article.

  1. What is the analysis used? Not mentioned in a section. 

Dear reviewer

We have made drastic corrections.

  1. To analyse, suggest using mixed-factorial ANOVA to analyse the results and not report individual data.

Dear reviewer

Thank you for your information, we have added the content in the article.

  1. Unfortunately, the discussion was not based on the results. Please add more linkage from the finding. 

Dear reviewer

We have made corrections.

  1. I felt the study need stronger justification, conceptual framework etc. By missing so much information, it creates more doubts for the reader than answers.

Dear reviewer

Regarding the research framework and related discussions, we have made substantial revisions. We believe in making articles more visual.

  1. References – please follow MDPI format. 

Dear reviewer

We have made corrections.

If there are still discrepancies, we will continue to make corrections.

The blue font is for corrections or supplements.

Thank you for the information, we have made a drastic correction. We believe in making articles more visual.

If there are still discrepancies, we will continue to make corrections.

Round 2

Reviewer 1 Report

Thanks for your corrections.

Now the paper is suitable for publication!

Reviewer 2 Report

I think the current manuscript read better and the authors had completed all the comments as suggested. Thus, I suggest accepting the manuscript with the condition to submit the manuscript for English editing and use IJERPH formatting. Thank you.